# Mining transcriptomic data to study the origins and evolution of a plant allopolyploid complex

Aureliano Bombarely[1], Jeremy E. Coate[2] and Jeff J. Doyle[1]

[1] Department of Plant Biology, Cornell University, Ithaca, NY, USA
[2] Department of Biology, Reed College, Portland, OR, USA

## ABSTRACT

Allopolyploidy combines two progenitor genomes in the same nucleus. It is a common speciation process, especially in plants. Deciphering the origins of polyploid species is a complex problem due to, among other things, extinct progenitors, multiple origins, gene flow between different polyploid populations, and loss of parental contributions through gene or chromosome loss. Among the perennial species of *Glycine*, the plant genus that includes the cultivated soybean (*G. max*), are eight allopolyploid species, three of which are studied here. Previous crossing studies and molecular systematic results from two nuclear gene sequences led to hypotheses of origin for these species from among extant diploid species. We use several phylogenetic and population genomics approaches to clarify the origins of the genomes of three of these allopolyploid species using single nucleotide polymorphism data and a guided transcriptome assembly. The results support the hypothesis that all three polyploid species are fixed hybrids combining the genomes of the two putative parents hypothesized on the basis of previous work. Based on mapping to the soybean reference genome, there appear to be no large regions for which one homoeologous contribution is missing. Phylogenetic analyses of 27 selected transcripts using a coalescent approach also are consistent with multiple origins for these allopolyploid species, and suggest that origins occurred within the last several hundred thousand years.

## INTRODUCTION

Polyploidy (whole genome duplication, WGD) is a key process in plant evolution. All seed plants are fundamentally polyploid, with a second WGD event shared by all flowering plants (*Jiao et al., 2011*), and additional events found in many lineages (see http://genomevolution.org/wiki/index.php/Plant_paleopolyploidy) (*Soltis et al., 2009*). It has been estimated that 15% of all flowering plant speciation events involve polyploidy (*Wood et al., 2009*). Systematists generally recognize autopolyploidy and allopolyploidy as distinct types of polyloidy events, based on the level of divergence of the diploid genomes that formed the polyploid. The terms are best thought of as describing elements of a continuum that ranges from the doubling of a single genome (autopolyploidy), to the

Corresponding author
Aureliano Bombarely,
ab782@cornell.edu

incorporation of differentiated genomes in a single nucleus by hybridization of different species (allopolyploidy). From a genetic perspective, allopolyploids are characterized by diploid-like meiotic behavior and limited interaction between the two homoeologous genomes. The duplicated chromosomes of an autopolyploid (and, to a lesser extent, a newly formed allopolyploid; *Ramsey & Schemske, 2002*) initially can associate randomly, leading to polysomic segregation, but it is generally assumed that this is a transient state; diploidization leads to the eventual presence of homoeologous genomes. It is difficult, if not impossible to determine from the genomes of older polyploids (paleopolyploids, mesopolyploids) how differentiated their progenitor genomes were in large part due to the frequent absence of extant diploid progenitors for comparative purposes.

The initial "fixed hybrid" condition of an allopolyploid erodes over time as homoeologous loci are lost (*Lynch & Conery, 2000*; *Maere et al., 2005*); this process of "fractionation" is thought to occur preferentially from one subgenome, but the precise mechanisms remain unknown (*Schnable & Freeling, 2011*; *Freeling et al., 2012*). In addition to the loss of genes, the process of concerted evolution can result in the replacement of a gene from one genome by its homoeologue, notably through gene conversion (e.g., *Wang et al., 2007*). The earliest stages of polyploid evolution may contribute disproportionately to gene loss and genomic rearrangement through genomic shock (*McClintock, 1984*). For example, some individuals of the ca. 100 year-old allopolyploid, *Tragopogon miscellus*, have lost entire chromosomes of one parent (*Chester et al., 2012*). Diversity in polyploids can be due to mutational divergence from parental diploids, but also due to multiple origins produced by different polyploidization events between different genotypes of the same diploid species (*Symonds, Soltis & Soltis, 2010*). Questions concerning how polyploids originate (e.g., single vs. multiple origins), how they partition their variation (e.g., as a single lineage united by gene flow vs. as separate lineages formed from different genotypes of the same progenitor species), and how much of the initial parental contributions they retain are among the major questions in polyploid evolutionary research (*Soltis et al., 2010*).

High-throughput sequencing produces massive amounts of genome-wide data, and thus has great potential for systematic and evolutionary studies in general (*Gilad, Pritchard & Thornton, 2009*). The ready availability of genomic and transcriptomic data has opened new opportunities for studying the evolution of polyploids (*Bombarely et al., 2012*; *Grover, Salmon & Wendel, 2012*; *Ilut et al., 2012*; *Dufresne et al., 2014*) at the scale of whole genomes. However, it is not trivial to extract relevant information from short read sequencing data, particularly for allopolyploids, where the interest is often in deconvoluting the complex genome into its two homoeologous subgenomes (*Grover, Salmon & Wendel, 2012*; *Ilut et al., 2012*). Moreover, the field of systematics has what has been called a new paradigm for studying species relationships, involving genealogical approaches (*Edwards, 2009*). Genealogical methods have lately begun to be applied to both autopolyploids (*Arnold, Bomblies & Wakeley, 2012*; *Hollister et al., 2012*) and allopolyploids (e.g., *Slotte et al., 2011*; *Jones, Sagitov & Oxelman, 2013*; *Slotte et al., 2013*). The confluence of these two developments promises to accelerate the study of polyploid evolution.

The genus *Glycine* includes the cultivated soybean (*G. max*) and its wild progenitor (*G. soja*), both annual species native to northeastern Asia, as well as approximately 30 perennial species native to Australia classified as subgenus *Glycine* (*Ratnaparkhe, Singh & Doyle, 2011*). Like many plant species, *Glycine* has a complex history of polyploidy: in addition to events shared with all angiosperms (*Jiao et al., 2011*) and eudicots (*Jiao et al., 2012*), the soybean genome retains evidence from a WGD around 50 million years ago (MYA) shared with a large subset of legumes (*Blanc & Wolfe, 2004*; *Schlueter et al., 2004*; *Cannon et al., 2010*), and particularly from a more recent polyploidy event that increased the chromosome number of the ancestor of all extant *Glycine* species from $2n = 20$ to $2n = 40$ (*Shoemaker, Schlueter & Doyle, 2006*; *Doyle & Egan, 2010*; *Schmutz et al., 2010*; *Doyle, 2012*). This *Glycine*-specific WGD occurred between the estimated time of homoeologous gene divergence in the soybean genome (10–13 MYA; e.g., *Egan & Doyle, 2010*; *Schmutz et al., 2010*), and around 5 MYA, when the annual and perennial species diverged from an already-polyploid common ancestor (*Doyle & Egan, 2010*).

In addition to these older events, eight perennial *Glycine* species are allopolyploids with $2n = 78$ or $80$, hypothesized to have arisen by hybridization involving various combinations of eight extant diploid species, several of them multiple times and involving both progenitors as chloroplast genome donors (*Doyle et al., 2004*). Various lines of evidence culminated in these hypotheses of reticulate relationships, which are shown in Fig. 1 for the six species that are part of the *G. tomentella* sub-complex (*Doyle et al., 2004*). Chromosome number polymorphism ($2n = 38, 40, 78, 80$) was observed in what was initially considered a single taxon, *Glycine tomentella* (*Newell & Hymowitz, 1978*). Patterns of sterility and partial chromosome pairing in artificial crosses among *G. tomentella* plants were consistent with the presence of shared homoeologous diploid genomes among polyploids (*Grant, Brown & Grace, 1984*; *Doyle et al., 1986*; *Singh, Kollipara & Hymowitz, 1998*). Isozyme studies of diploid and allopolyploid *G. tomentella* led to the characterization of numerous "races" designated either "D" for diploid, or "T" for tetraploid (*Doyle & Brown, 1985*; *Singh, Kollipara & Hymowitz, 1998*). Morphological complexity, presumably due to the reticulate nature of the complex, has slowed nomenclatural recognition of what are clearly species in the biological sense. More recently, molecular phylogenetic studies assumed a dominant role in refining hypotheses of relationships (*Hsing et al., 2001*; *Singh, Kollipara & Hymowitz, 1998*; *Brown et al., 2002*; *Doyle et al., 2002*; *Rauscher, Doyle & Brown, 2004*), and corroborated earlier hypotheses concerning the origins of polyploids from among the diploid ($2n = 38, 40$) "genome groups" that were also initially defined by artificial hybridization studies and later by molecular studies (see *Ratnaparkhe, Singh & Doyle, 2011*). However, these DNA-level studies were based on only two molecular markers: the internal transcribed spacers of the 18S-5.8S-26S nuclear ribosomal gene cistron (nrDNA ITS) and the low copy nuclear gene, histone H3D. Relationships of chloroplast genomes are broadly consistent with these results (*Hsing et al., 2001*), but are complicated by incongruence with nuclear markers, likely due to a combination of incomplete lineage sorting and introgression (*Doyle et al., 2004*). Thus, a genome-wide perspective on the

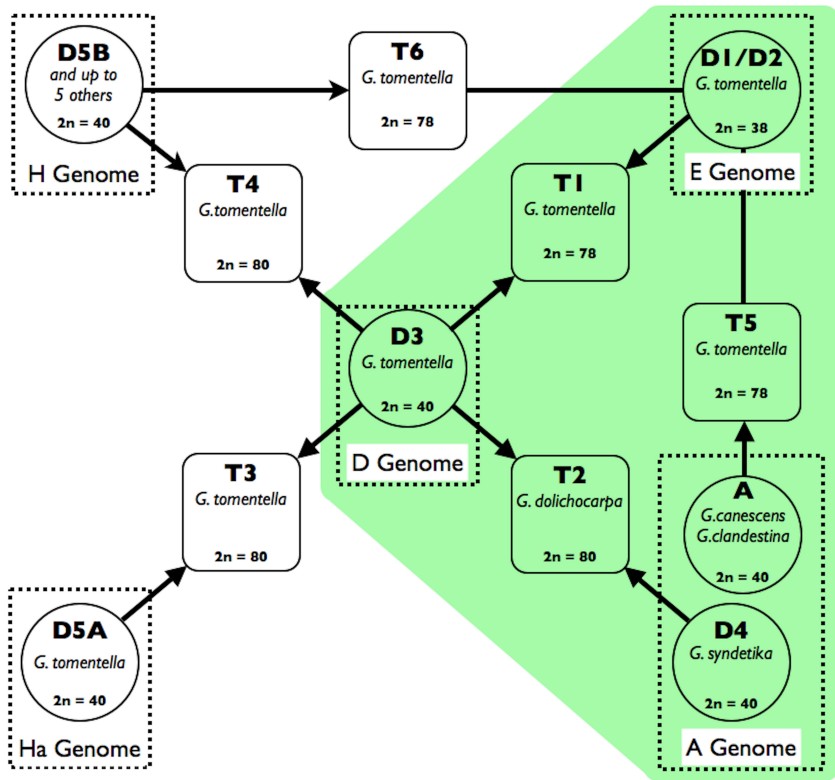

**Figure 1 Schema of the *Glycine* perennial polyploid complex.** Diploid progenitors are represented by circles and allotetraploid species by squares. Chromosome numbers are shown for each species, and genome groups (*Ratnaparkhe, Singh & Doyle, 2011*) are given for diploids. Species used in this study (*G. tomentella* D1, *G. tomentella* D3, *G. syndetika* D4, *G. canescens*, *G. clandestina*, *G. dolichocarpa* T2, *G. tomentella* T1 and *G. tomentella* T5) are shown in green.

origin and evolution of the *G. tomentella* complex, including estimates of dates of origin, has been lacking.

A better understanding of the origin and evolution of the *Glycine* allopolyploid complex will complement its exploitation in studying the impact of allopolyploidy on a range of morphological and physiological characters (*Coate & Doyle, 2010*; *Coate et al., 2012*; *Ilut et al., 2012*; *Coate & Doyle, 2013*; *Hegarty et al., 2013*). Here we apply phylogenetic and coalescent methods to a transcriptomic dataset from three of these allopolyploid species and their diploid progenitors that was originally generated to study the effects of polyploidy on their ability to cope with stress from excess light (*Coate & Doyle, 2013*).

## MATERIAL AND METHODS

### Taxon sampling and transcriptome sequencing

Three *Glycine* (Fig. 1) allopolyploid "triads" (from the *Glycine* perennial polyploid complex) defined as an allopolyploid species and its two putative diploid progenitors, were sampled: (1) the allopolyploid, *G. tomentella* T1 ($2n = 78$) and the diploid species, *G. tomentella* D1 (E-genome of *Hymowitz, Singh & Kollipara, 2010*; $2n = 38$) and *G. tomentella* D3 (D-genome; $2n = 40$); (2) *G. dolichocarpa* (= *G. tomentella* T2; $2n = 80$)

**Table 1 Sequencing, reads processing and mapping summary.** Represented genes reflected the number of *Glycine max* reference genome genes where after the perennials reads mapping and expression measure have an expression >0 (RPKM). Gray shading, allopolyploid species.

| Species | Accession | Samples | Raw reads | Processed reads | Mapped reads | Represented genes |
|---|---|---|---|---|---|---|
| *Glycine canescens* | 1232 | 2 | 21,332,880 | 20,696,801 | 14,381,555 | 23,833 |
| *Glycine clandestina* | 1126 | 2 | 19,086,864 | 18,613,018 | 11,815,996 | 23,340 |
| | 1253 | 3 | 33,546,015 | 32,326,942 | 19,117,095 | 23,723 |
| *Glycine dolichocarpa* | 1134 | 13 | 202,427,873 | 187,120,918 | 60,712,525 | 23,643 |
| | 1188 | 2 | 19,034,633 | 18,279,858 | 11,960,713 | 22,952 |
| | 1286 | 2 | 11,814,995 | 11,216,980 | 7,422,888 | 25,278 |
| | 1393 | 2 | 21,820,163 | 21,029,983 | 13,643,602 | 23,345 |
| | 1854 | 3 | 54,748,079 | 42,826,840 | 16,032,643 | 22,718 |
| *Glycine syndetika* | 1300 | 3 | 25,527,322 | 23,634,740 | 14,092,961 | 24,238 |
| | 2073 | 2 | 12,132,989 | 11,072,073 | 7,087,710 | 24,438 |
| | 2321 | 2 | 32,796,391 | 30,024,544 | 13,637,368 | 22,571 |
| *Glycine tomentella* D1 | 1156 | 3 | 38,218,179 | 36,988,846 | 21,905,536 | 23,041 |
| | 1157 | 2 | 16,522,541 | 15,906,072 | 9,715,890 | 23,920 |
| | 1316 | 2 | 25,207,045 | 24,375,482 | 15,417,078 | 22,749 |
| *Glycine tomentella* D3 | 1364 | 1 | 10,401,944 | 9,604,350 | 6,896,983 | 22,802 |
| | 1366 | 2 | 20,631,583 | 18,098,232 | 10,766,169 | 23,364 |
| | 1403 | 3 | 31,631,369 | 28,953,234 | 17,218,424 | 23,352 |
| | 1820 | 3 | 71,185,274 | 63,055,644 | 18,625,439 | 22,871 |
| *Glycine tomentella* T1 | 1288 | 2 | 14,608,219 | 14,148,847 | 9,298,349 | 23,348 |
| | 1361 | 2 | 17,964,870 | 17,627,119 | 11,217,736 | 23,758 |
| | 1763 | 2 | 21,870,236 | 20,933,661 | 14,101,838 | 23,349 |
| *Glycine tomentella* T5 | A58_1 | 2 | 22,447,334 | 21,996,303 | 13,389,955 | 23,042 |
| | 1487 | 2 | 21,267,274 | 20,469,069 | 13,907,305 | 23,437 |
| | 1969 | 3 | 21,324,229 | 20,847,883 | 11,136,293 | 23,522 |

and its putative progenitors *G. tomentella* D3 and *G. syndetika* (= *G. tomentella* D4; A-genome; $2n = 40$); and (3) *G. tomentella* T5 ($2n = 78$) and its hypothesized progenitors, *G. tomentella* D1 and *G. clandestina* (A-genome; $2n = 40$). Each species was represented by 2–5 accessions sampled from the CSIRO Division of Plant Industry Perennial *Glycine* Germplasm Collection (Table 1). Additionally, a synthetic allotetraploid (A58) was used, which mimics the natural T5 allopolyploid, having been produced by doubling an artificial hybrid of *G. tomentella* D1 (accession G1316) and *G. canescens* (accession G1233; A-genome; $2n = 40$); *G. canescens* is an A-genome species closely related to *G. clandestina*. A summary of the datasets used can be found in Table S4.

Plants were grown in a common growth chamber with a 12 h/12 h light/dark cycle, 22 °C/18 °C day/night temperature regime, and a light intensity of either 125 mmol $m^{-2}$ $s^{-1}$ (LL) or 800 mmol $m^{-2}$ $s^{-1}$ (EL). Different light intensities were used for the purposes of a separate study examining light stress responses (*Coate & Doyle, 2013*). Single leaflets were pooled from six individuals per accession, and RNA-Seq libraries were constructed from the pooled tissue. All samples were taken from approximately 1-week-old, fully expanded leaves, and were collected 0.5–2.0 h into the light period.

For each light treatment, all tissue was collected in a single morning and immediately frozen in liquid nitrogen. Total RNA was isolated from pooled leaf tissue using the Plant RNeasy Kit with on-column DNase treatment (Qiagen, Valencia, CA, USA). Single-end RNA-Seq libraries were constructed following the Illumina mRNA-seq Sample Preparation Kit protocol (Illumina, San Diego, CA, USA), with the following modifications: (1) two rounds of polyA selection were performed using the Dynabeads mRNA DIRECT Kit (Life Technologies, Carlsbad, CA, USA); (2) RNA was fragmented for 2 min at 70 °C using the RNA fragmentation reagents kit (Life Technologies); and (3) Illumina PE adapters were replaced with custom-made adapters containing 3nt barcodes in order to facilitate multiplexing of samples (see *Coate & Doyle, 2013* for adapters and Table S1 for the barcode sequences). Sequencing was performed on either the GAIIx or HiSeq 2000 platform (Illumina), generating 88 nt or 100 nt reads, respectively. Equimolar amounts of three (GAIIx) or four (HiSeq 2000) barcoded libraries were combined and sequenced per channel.

## Read processing and single nucleotide polymorphism (SNP) calling

Reads were processed with Fastq-mcf (*Aronesty, 2013*) to trim low quality extremes (min. quality 30) and remove short reads (min. read length 50 bp). They were aligned to the soybean genome (version 1.0, downloaded from www.phytozome.net/soybean) using Bowtie2 (*Langmead & Salzberg, 2012*) with the default parameters. Mapping files from the same accession were merged. Reads without preferential mapping (same score for two or more mapping hits) and with a mapping score below 20 were removed. SNP calling was performed using Samtools (*Li et al., 2009*). SNPs supported with read coverage below 5 were removed. VCF files were combined and formatted to Structure and Hapmap formats using the Perl script MultiVcfTool (https://github.com/aubombarely/GenoToolBox/blob/master/SeqTools/MultiVcfTool).

## Homoeologue read identification and transcript-guided assembly

For homoeologous SNP identification, a consensus diploid transcriptome was rebuilt for each of the species groups (A, with *G. clandestina* and *G. canescens* accessions; D1, with *G. tomentella* D1 accessions; D3, with *G. tomentella* D3 accessions; and D4, with *G. syndetika* accessions) using Samtools (*Li et al., 2009*) and Gffread from the Cufflinks software package (*Trapnell et al., 2010*). A progenitor reference set was created for each of the polyploid species joining the diploid transcriptome sets (T1 = D1 + D3, T2 = D3 + D4 and T5 = A + D1). Reads from the polyploid species were mapped with these references using Bowtie2. Sam mapping files were processed to identify reads according the preferential mapping with each of the progenitors using the Perl script, SeparateHomeolog2Sam (https://github.com/aubombarely/GenoToolBox/blob/master/SeqTools/SeparateHomeolog2Sam). Reads with mapping score AS and XS = 0 (No SNPs) were kept and used to rebuild the polyploid transcriptomes using Samtools (*Li et al., 2009*) and Gffread (from the Cufflinks package *Trapnell et al., 2012*). Once the reads were separated according its preferential mapping, they were mapped back to the soybean genome. SNPs were called as described above.

## Population structure analysis

The programs Structure (*Pritchard, Stephens & Donnelly, 2000*) and fineStructure (*Lawson et al., 2012*) were used to analyze population structure of the two SNP datasets, with and without polyploid SNPs separated by homoeologue, described above. For Structure, each of the datasets was divided into three subsets of 20,000 SNPs selected with a random function incorporated in the MultiVcfTool. 5 replicates were run for each of the subsets with a burn-in of 10,000 and a number of MCMC repetitions of 10,000, from $K = 1$ to $K = 15$ using the default parameters ($\lambda = 1$, assuming uniform distribution of allele frequencies, *Pritchard, Stephens & Donnelly, 2000*). Admixture was selected. The optimal number of clusters was identified based on the rate of change in the log probability of data between successive $K$ values (*Evanno, Regnaut & Goudet, 2005*). Results at $K = 6$ were verified with a re-analysis using a burn-in of 100,000 generations. Results were visualized using R (barplot function).

The two SNP datasets were divided into 20 different subsets each mapping to one soybean reference chromosome for FineStructure analysis. Analyses were performed following the instructions from the fineStructure web for the unlinked model (http://www.maths.bris.ac.uk/~madjl/finestructure/data_example.html). Results were presented as a heatmap of distances between each of the accessions. A principal component analysis (PCA) was performed over the same distance matrix using fineStructure software. The PCA figure was created using R.

## Reconstruction of phylogenies using concatenated SNPs

SNPs from the dataset in which SNPs from allopolyploids were partitioned into their two homoeologues ("homoeologue data set") and were concatenated to create a supermatrix with 36 operational taxonomic units (OTUs). The two homoeologous gene copies from each allopolyploid were treated as individual OTUs; for example the D1 and D3 homoeologues of T1 individuals were treated as D1T1 and D3T1, respectively. *G. max*, accession William82 was used as outgroup. The alignment files were produced changing the SNPs Hapmap format to fasta using a Perl script. The resulting matrix was used in two analyses. First, maximum likelihood (ML) was used, implemented in PhyML (*Guindon & Gascuel, 2003*) with GTR as the substitution model; 100 bootstrap replicates were conducted. Second, in order to visualize reticulations in the dataset, a network method, NeighborNet, was implemented in the SplitsTree package (*Huson & Bryant, 2006*) with the default parameters. Trees were visualized and drawn using FigTree (*Rambaut, 2012*).

## Gene-based analyses

A subset of transcripts was selected for phylogenetic and network analyses based on the following criteria: No more than 10% of Ns for the guided assembly consensus sequence in any of the accessions after the homoeologue read identification; alignments with at least 1000 bp; and genes with their corresponding *G. max* homologue identified as an existing pair retained from the most recent (ca. 5–10 million years; *Doyle & Egan, 2010*) *Glycine* WGD event, as compiled by *Du et al. (2012)*. Sequence alignments were based on

the transcriptome-guided assembly. Sequence for each of the genes was collected with a Perl script (FastaSeqExtract, GenoToolBox script package), concatenated and changed to the required sequence alignment format using a BioPerl script (bp_sreformat.pl). The 95 alignments selected were used in an exploratory phylogenetic analysis using the Bayesian MCMC method, BEAST (*Drummond et al., 2012*) (HKY substitution model, 10,000,000 MCMC). Alignments that produced trees in which *G. max* was not sister to perennial *Glycine* species in the consensus tree were removed. Generally the removed alignments showed tree topologies with two large clades with long branches, indicating the possibility of inclusion of paralogous genes from the older whole genome duplication (ca. 50 MY, common to the Leguminosae; reviewed in *Doyle, 2012*) instead the orthologue.

27 genes selected after this filtering were analyzed using three different methods: (1) Phylogenies were reconstructed using ML using PhyML (*Guindon & Gascuel, 2003*) with 1,000 bootstraps. jModelTest2 was used to choose the best substitution model (*Darriba et al., 2012*). According to the Bayesian Information Criterion (BIC) HKY was the preferred model (40% of the genes), followed by K80 (26% of the genes; Table S2). (2) Networks were constructed using NeighborNet in SplitsTree4 with the default parameters (*Huson & Bryant, 2006*). (3) Bayesian analysis was performed using BEAST v2.0 (*Drummond et al., 2012*). The two homoeologous gene copies from each allopolyploid were treated as individual OTUs as in the concatenated analysis, and *G. max*, accession William82 was again used as outgroup. Based on the jModelTest2 results, HKY was used as the substitution model. The MCMC chain was set to 100,000,000 MCMC generations, taking samples every 1000 generations. Divergence ages were estimated by scaling the tree root (divergence between *G. max* and perennials) to 5 Myr (*Egan & Doyle, 2010*). All trees were drawn using FigTree (*Rambaut, 2012*).

## Species tree reconstruction

Species tree reconstruction under the coalescent was performed using the 27 selected genes in *BEAST (*Drummond et al., 2012*). The 24 accessions, including two homoeologues for each allopolyploid accession, were grouped in 11 operational taxonomic units (OTUs) for this analysis: *G. canescens, G. clandestina, G. tomentella* D1, *G. tomentella* D3, *G. syndetika* (D4), *G. tomentella* T1–D1, *G. tomentella* T1–D3, *G. dolichocarpa* T2–D3, *G. dolichocarpa* T2–D4, *G. tomentella* T5–A *and G. tomentella* T5–D1. *G. max* was used as outgroup. Based on jModelTest2 results, HKY was used as substitution model. The MCMC chain was set to 100,000,000 MCMC generations, taking samples every 1000 generations. Divergence dates were estimated as described above. All the trees were drawn using FigTree (*Rambaut, 2012*).

## RESULTS

### Phylogenomics dataset generation

Between 7 and 60 million reads from leaf transcriptomes of 24 accessions representing 8 *Glycine* perennial species were mapped to the *Glycine max* genome (v1.0) (*Schmutz et al., 2010*). Reads mapped to 22,500–25,000 genes (∼40% of soybean gene models; Table 1); this represents between 4.5 and 11.6% of the genome. 200,000–965,000 single nucleotide

**Table 2  Summary of SNPs using *G. max* as reference genome.** Gray shading, allopolyploid species.

| Species | Accession | % Gmax coverage[*] | Raw SNPs | Processed SNPs[**] | Synonymous | Non-synonymous |
|---|---|---|---|---|---|---|
| *Glycine canescens* | 1232 | 7.2 [65.0] | 589,686 | 453,398 [7.7] | 148,321 | 123,413 |
| *Glycine clandestina* | 1126 | 6.7 [61.4] | 496,746 | 375,943 [7.5] | 115,340 | 96,562 |
| | 1253 | 7.5 [65.1] | 617,543 | 487,923 [8.3] | 143,952 | 124,920 |
| *Glycine dolichocarpa* | 1134 | 11.6 [77.4] | 1,135,676 | 965,643 [26.4] | 242,556 | 221,326 |
| | 1188 | 7.4 [65.1] | 550,698 | 423,353 [28.9] | 132,471 | 113,785 |
| | 1286 | 4.5 [45.5] | 302,661 | 224,653 [27.9] | 67,187 | 53,595 |
| | 1393 | 6.7 [62.9] | 470,402 | 367,646 [28.8] | 125,339 | 104,549 |
| | 1854 | 7.8 [65.3] | 580,531 | 471,020 [25.8] | 140,911 | 120,274 |
| *Glycine syndetika* | 1300 | 7.5 [65.5] | 605,556 | 477,245 [7.8] | 147,362 | 125,041 |
| | 2073 | 6.0 [57.6] | 402,798 | 282,215 [12.6] | 91,451 | 75,333 |
| | 2321 | 8.0 [67.7] | 670,121 | 544,101 [6.3] | 166,409 | 143,612 |
| *Glycine tomentella* D1 | 1156 | 8.6 [69.7] | 767,614 | 621,043 [7.6] | 190,778 | 160,781 |
| | 1157 | 6.2 [56.8] | 455,265 | 328,574 [7.1] | 94,377 | 77,945 |
| | 1316 | 7.2 [62.3] | 537,439 | 412,518 [9.3] | 120,056 | 99,666 |
| *Glycine tomentella* D3 | 1364 | 5.0 [51.8] | 335,301 | 226,697 [7.8] | 84,917 | 65,888 |
| | 1366 | 6.6 [59.7] | 481,258 | 360,327 [7.5] | 111,011 | 90,015 |
| | 1403 | 6.4 [60.8] | 476,495 | 369,661 [6.6] | 121,526 | 99,074 |
| | 1820 | 9.3 [69.6] | 803,774 | 641,145 [6.6] | 188,965 | 161,826 |
| *Glycine tomentella* T1 | 1288 | 6.9 [63.0] | 498,900 | 371,845 [19.6] | 121,418 | 102,548 |
| | 1361 | 5.1 [54.2] | 293,339 | 200,738 [18.4] | 75,653 | 59,378 |
| | 1763 | 7.1 [65.5] | 533,041 | 417,420 [19.4] | 140,203 | 116,465 |
| *Glycine tomentella* T5 | A58_1 | 7.3 [64.6] | 544,331 | 430,552 [27.3] | 135,163 | 113,781 |
| | 1487 | 7.0 [63.6] | 516,755 | 395,503 [26.8] | 128,199 | 105,647 |
| | 1969 | 7.4 [65.9] | 558,920 | 444,468 [27.5] | 146,711 | 124,933 |

**Notes.**

[*] Between square brackets the coverage of the *G. max* transcriptome, including alternative splicings.

[**] Square brackets, percentage of heterozygous positions.

polymorphisms (SNPs) were identified relative to *G. max*; 6.3–12.6% of SNP positions were polymorphic in diploid species (*G. clandestina*, *G. canescens*, *G. tomentella* D1 (referred as D1 hereafter), *G. tomentella* D3 (referred as D3) and *G. syndetika* (referred as D4)), and 18.4–28.8% in polyploid species (*G. tomentella* T1 (referred as T1), *G. tomentella* T5 (referred as T5) and *G. dolichocarpa* T2 (referred as T2); Table 2). The interpretation of these positions as standard heterozygosity is complicated by the recent (5–10 MYA: *Doyle & Egan, 2010*) WGD in the ancestral *Glycine* genome. In a gene for which soybean has lost one of the homoeologous copies from this event, but the perennial species for which it is serving as reference has retained both copies, polymorphic SNPs may be due to reads from two different homoeologous loci in the perennial, rather than two alleles at a single locus. Low levels of conventional heterozygosity are expected in *Glycine* species, because of their strongly selfing reproductive biology, with much reproduction occurring through cleistogamous (closed, selfing) flowers.
**Table 3 Summary of the mapped reads and SNPs produced after the homoeologus reads separation.** It is based in the selective mapping with its progenitors.

| Species | Accession | Progenitor I | Mapped to progenitor I (%) | SNPs for I[*] | Progenitor II | Mapped to progenitor II (%) | SNPs for II[*] |
|---|---|---|---|---|---|---|---|
| *Glycine dolichocarpa* | 1134 | D3 | 11.4 | 399,884 [2.2] | D4 | 11.6 | 380,389 [2.1] |
| | 1188 | D3 | 20.8 | 227,610 [2.0] | D4 | 20.4 | 220,610 [2.1] |
| | 1286 | D3 | 20.3 | 124,984 [1.7] | D4 | 20.3 | 123,873 [1.8] |
| | 1393 | D3 | 19.6 | 197,132 [1.9] | D4 | 19.8 | 192,148 [1.9] |
| | 1854 | D3 | 17.9 | 245354 [1.5] | D4 | 19.3 | 242,561 [1.7] |
| *Glycine tomentella T1* | 1288 | D1 | 14.9 | 143,232[1.7] | D3 | 17.5 | 160,873 [1.9] |
| | 1361 | D1 | 15.0 | 155,360 [1.6] | D3 | 17.6 | 175,871 [2.0] |
| | 1763 | D1 | 14.8 | 158,777 [1.8] | D3 | 17.3 | 179,032 [2.0] |
| *Glycine tomentella T5* | A58_1 | A | 16.9 | 190,138[2.1] | D1 | 20.5 | 222,134 [1.8] |
| | 1487 | A | 17.1 | 174,051 [1.9] | D1 | 20.0 | 202,555 [1.7] |
| | 1969 | A | 16.0 | 182,615 [2.4] | D1 | 18.6 | 214,799 [1.8] |

**Notes.**

[*] Square brackets, percentage of heterozygous positions.

The much higher percentage of polymorphic positions in polyploid individuals (T1, T2, T5) likely is also due to the mapping of reads from two homoeologous copies to a single target, in this case due to recent polyploidy: for example, mapping reads from tetraploid ($2n = 80$) T2 to a single locus in the diploid ($2n = 40$) *G. max* reference genome will result in reads from both its D3 and D4 homoeologous subgenomes mapping to the same target, increasing the chance of observing a polymorphism at a given site. Separating reads from T1, T2, and T5 polyploid individuals was possible where the read has at least one SNP that could be related to one homoeologous genome contributor (e.g., D3 and D4 differed by a SNP) and this difference was retained in the D3 and D4 homoeologous genomes of T2; diploid-distinguishing polymorphism (DDP; see *Ilut et al., 2012*). Between 11.4 and 20.8% of reads were assigned to one of the progenitors (Table 3).

Between 124,984 and 399,884 SNPs were produced for each accession. The filtering of the missing data produced 237,243 and 75,958 polymorphic positions for all the accessions before and after the homoeologous read assignment, respectively. SNPs per chromosome ranged from 7,455 (chromosome 14) to 16,494 (chromosome 8) and from 2,288 (chromosome 14) to 5,300 (chromosome 8) before and after the homoeologous read assignment, respectively. SNPs per species group ranged from 21,830 (D1 species) to 26,438 (A species, *G. canescens* and *G. clandestina*) (Table 4).

Transcriptome-guided assemblies produced between ~1,800 and ~6,600 full-length sequences (as mapped to the *G. max* gene models) for each diploid accession. For polyploid subtranscriptomes this number was much lower because only reads that mapped preferentially to one of the diploid consensus species and reads that mapped equally but with no polymorphism (perfect match) were used during the transcriptome-guided assembly. Any read that mapped equally to two or more positions with one or more polymorphisms was discarded because it was impossible to assign it to any of the diploid

**Table 4 Summary of SNP count between species groups.** Polyploids are divided in two species according the progenitor origin. A Species includes *G. canescens*, *G. clandestina* and *G. tomentella* T5–A; D1 species includes *G. tomentella* D1, *G. tomentella* T1–D1 and *G. tomentella* T5–D1; D3 species includes *G. tomentella* D3, *G. tomentella* T1–D3 and *G. tomentella* T2–D3; D4 species includes *G. syndetika* and *G. tomentella* T2–D4.

| Species group | *Gmax* SNPs | A group SNPs | D1 group SNPs | D3 group SNPs | D4 group SNPs |
|---|---|---|---|---|---|
| A species | 9,406 | 26,438[*] | 7,096 | 6,591 | 1,465 |
| D1 species | 11,187 | – | 21,830[*] | 5,933 | 7,556 |
| D3 species | 9,299 | – | – | 25,157[*] | 7,295 |
| D4 species | 9,314 | – | – | – | 23,324[*] |

**Notes.**

[*] The same species group contains the specific SNPs between accession of the same species.

progenitors, reducing the mapping coverage of the reference gene models. Between ∼350 and ∼1,350 full length sequences were assembled for the T1, T2, and T5 polyploid homoeologous subtranscriptomes of which between 4 and 19% were duplicated genes from the 5 to 10 MYA WGD event in the common ancestor of *Glycine* species (*Schmutz et al., 2010*). For phylogenetic analysis, full length sequences are not needed so a phylogenetic analysis dataset was created with 27 genes (see Material and Methods for the criteria used to generate this dataset; Table 5).

## Genome-wide distribution of homoeologous SNPs

For each allopolyploid accession, the ca. 120,000–400,000 SNPs (Table 3) that could be identified to the homoeologous subgenome were mapped to the soybean reference genome (*Schmutz et al., 2010*). This produced a map that is analogous to chromosome painting (genomic in situ hybridization, GISH) experiments using the reads from which the SNPs were derived, which we term "electronic chromosome painting" (e-chromosome painting). Similar patterns were seen for all accessions, with high densities of SNPs at the ends of each soybean chromosome and far lower densities in pericentromeric regions (Fig. 2). This pattern is expected using reads from transcriptome data, because of the sparse distribution of genes in pericentromeric regions of the soybean genome (*Schmutz et al., 2010*). Notably, in all allopolyploid accessions, SNPs from both homoeologues were distributed across the entire genome, and no regions were identified in which SNPs from only one homoeologue were mapped (Fig. 2; Figs. S1–S10).

## Population structure analyses

Structure (*Pritchard, Stephens & Donnelly, 2000*) was first run using all available SNPs, without separating SNPs from polyploids into homoeologous groups. Structure was run from $K = 1$–15; $K = 6$ was identified as one of the optimal preferred values of $K$ using the delta $K$ method of *Evanno, Regnaut & Goudet* (*2005*; Fig. S11). Five of these six groups corresponded to diploid taxa: D1, D3, D4, *G. canescens*, and *G. clandestina* (Fig. 3A). The sixth group was represented only as a minor component in D4 accession 2073. Diploid accessions showed little or no evidence of admixture, with the exception of D4 accession 2073 (Fig. 3). In contrast, all polyploid accessions were admixed, each with

**Table 5  Summary of the genes used in the BEAST and *BEAST analysis.** Tree likelihood values and the functional annotation are shown.

| GeneID | TreeLikelihood mean | TreeLikelihood ESS | Gene functional annotation |
| --- | --- | --- | --- |
| Glyma01g35620 | −4,676.067 | 1,361.926 | Phytoene dehydrogenase |
| Glyma02g11580 | −3,986.812 | 1,034.414 | RNA binding protein |
| Glyma03g29330 | −7,611.686 | 894.813 | Magnesium chelatase |
| Glyma03g36630 | −2,666.725 | 696.197 | Rho GTPase activating protein |
| Glyma04g39670 | −4,142.157 | 2,556.251 | ATP-binding transport protein-related |
| Glyma05g05750 | −3,028.809 | 541.483 | Beta-amylase |
| Glyma05g09310 | −2,578.198 | 305.191 | Pyruvate kinase |
| Glyma05g26230 | −3,741.498 | 5,156.337 | Metalloprotease M41 FtsH |
| Glyma05g37840 | −2,138.404 | 3,766.767 | Haloacid dehalogenase-like hydrolase |
| Glyma06g18640 | −3,418.91 | 6,613.752 | Elongation factor Tu |
| Glyma07g03370 | −2,091.845 | 742.796 | Palmytoil-monogalactosyldiacylglycerol delta-7 desaturase |
| Glyma07g17180 | −2,218.934 | 2,833.162 | Fructose-1,6-bisphosphatase |
| Glyma10g42100 | −2,903.849 | 1,774.192 | 3-ketoacyl-CoA synthase |
| Glyma11g13880 | −4,644.419 | 7,487.252 | Lipoxygenase |
| Glyma11g33720 | −3,592.689 | 2,273.389 | DELLA protein |
| Glyma12g04150 | −2,061.527 | 4,777.335 | Fructose-bisphosphate aldolase |
| Glyma12g12230 | −2,177.72 | 1,790.185 | O-methyltransferase |
| Glyma13g17820 | −2,439.715 | 342.999 | Polyubiquitin |
| Glyma14g03500 | −2,063.541 | 819.216 | Phytoene synthase |
| Glyma16g00410 | −4,041.294 | 4,475.536 | heat shock protein 70 |
| Glyma16g01980 | −4,985.489 | 387.993 | Myb-like protein |
| Glyma16g04940 | −2,152.355 | 4,586.42 | Glyceraldehyde 3-phosphate dehydrogenase |
| Glyma18g04080 | −2,285.776 | 9,535.754 | 26S proteasome regulatory complex, ATPase RPT4 |
| Glyma19g03390 | −2,344.831 | 3,190.5 | Unknown |
| Glyma19g32940 | −2,176.029 | 2,579.558 | Fatty acid desaturase |
| Glyma20g24930 | −2,803.585 | 6,535.602 | 3-ketoacyl-CoA synthase |
| Glyma20g32930 | −2,867.321 | 2,078.549 | Cytochrome P450 77A3 |

approximately 50% contributions from two different diploid groups. The genomic makeup of each accession was as expected from previous hypotheses (e.g., *Doyle et al., 2002*; Fig. 1): T1 accessions showed admixture from D1 and D3, T2 accessions from D3 and D4, and natural T5 accessions from D1 and *G. clandestina*. The synthetic T5 accession (A58) was also admixed, with contributions from D1 and *G. canescens*, as expected (*Joly et al., 2004*).

A second Structure analysis was conducted with each polyploid accession treated as two separate OTUs, using the homoeologue dataset (Table 2). As with the previous analysis, the analysis was run for $K = 1$–15. The Evanno method (*Evanno, Regnaut & Goudet, 2005*) identified $K = 6$ and 9 as the preferred values (Fig. S11). In the case of $K = 9$ the group representation shows the same structure than the $K = 6$ (Fig. S12). Results for diploids were similar to those obtained in the previous analysis (Fig. 3B). Subgenomes from natural allopolyploids and the synthetic T5 allopolyploid (A58) were shown to belong exclusively to diploid groups, with little or no evidence of admixture, indicating that the SNP filtering into homoeologous contributions was successful.

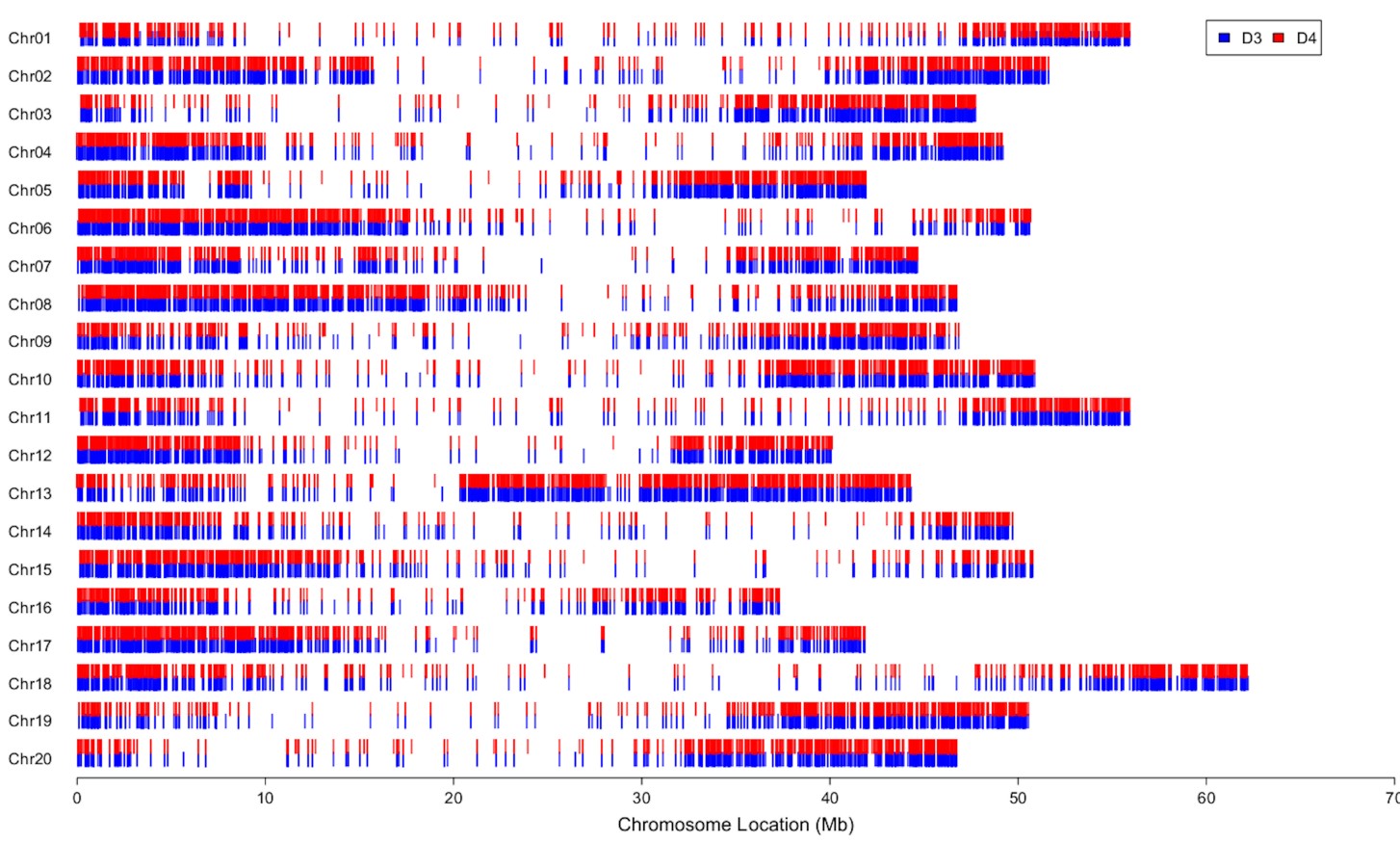

**Figure 2  Electronic chromosome painting for *G. dolichocarpa* T2 accession 1134.** SNP positions on the 20 soybean chromosomes are represented by blue lines (D3 progenitor) or red lines (D4 progenitor).

Complementary to the second Structure analysis, the data were analyzed using ChromoPainter and FineStructure (*Lawson et al., 2012*). ChromoPainter produces a co-ancestry matrix (as a measure of the ancestry sharing between individuals) based on the haplotype information provided by shared chunks (regions) of biallelic markers between individuals (*Lawson et al., 2012*). The two SNP datasets were filtered by selecting only the biallelic markers, producing a subset with 220,952 and 71,610 SNPs (before and after homoeologous read assignment, respectively) distributed along all 20 soybean chromosomes. Regions identified by ChromoPainter for each accession ranged from 516 (D4 2321) to 567 (*G. clandestina* 1253) and from 202 (D4 1300 and 2321) to 221 (D4 2073) (before and after homoeologous read assignment respectively). Principal component analysis (PCA) and population relationship analysis using a Bayesian approach were performed over the co-ancestry matrix using FineStructure (*Lawson et al., 2012*). PCA before homoeologous read assignment (Fig. 4A) shows seven well-differentiated groups, one per species with the exception that *G. canescens* and *G. clandestina* clustered together. Diploid species formed the vertices of a trapezoid. A-genome species (*G. canescens*, *G. clandestina* and D4) formed a more dispersed group than either D1 or D3. Each
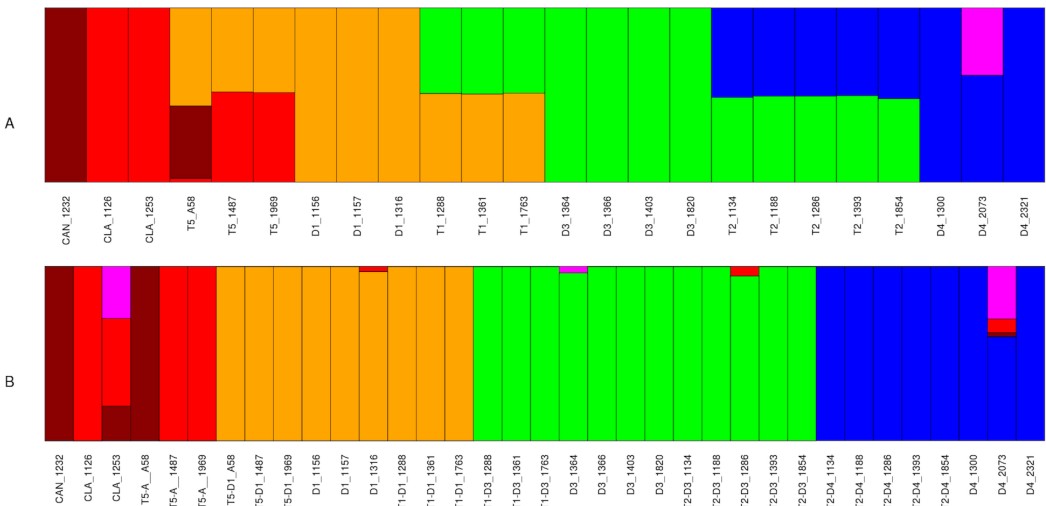

**Figure 3 Structure analysis for *Glycine* perennial polyploid accessions.** SNP analysis using Structure for a set of 20,000 random SNPs for *Glycine* polyploid complex accessions (A) without homoeologue separation and (B) with homoeologue separation for $K = 6$. The five progenitor diploid species are placed in different populations: red (*G. clandestina*), dark red (*G. canescens*), yellow (*G. tomentella* D1), blue (*G. syndetika* D4) and green (*G. tomentella* D3).

polyploid species fell between its putative diploid progenitors, consistent with each being an admixture (fixed hybrid). After the homoeologous read assignment (Fig. 4B), each of the polyploid subgenomes clustered with its diploid progenitors, producing three clear clusters: D1, D3, and A-genome (comprising *G. canescens, G. clandestina* and D4, as expected). Heatmaps were used to visualize the population relationships produced by FineStructure, complementing the information shown by the PCA figures. The heatmap before homoeologous read assignment (Fig. 4C), showed four intense regions (red, magenta and blue colors) corresponding to the four species groups of the PCA (Fig. 4A). Each polyploid showed the expected similarity to its progenitors; similarly, as expected the two *G. clandestina* accessions were more similar to one another than either was to *G. canescens*. Also, T5 A58, the artificial polyploid produced from a cross between *G. canescens* 1232 and D1 1316, showed the expected relationships with these accessions. Other T5 polyploids also showed a stronger signal from D1 1316 than from other D1 accessions. T2 accessions did not show any stronger signal with any particular D3 accession than with others, but they did with the D4 accessions 1300 and 2321, relative to 2073. T1 accessions 1288 and 1763 also showed a stronger signal with particular D1 and D3 accessions, whereas T1 accession 1361 showed a weaker signal with the D1 and D3 accessions included here. After the homoeologous read assignment (Fig. 4D), some of these signals were intensified, such as the relationship between T5 and D1 subgenomes and particular D1 accessions, but other relationships that were suggested when all SNPs were considered were not observed (for example there is not a stronger signal of D1 1316 with the T5 accessions). These differences may be due to the methodology used for the homoeologous read assignment.

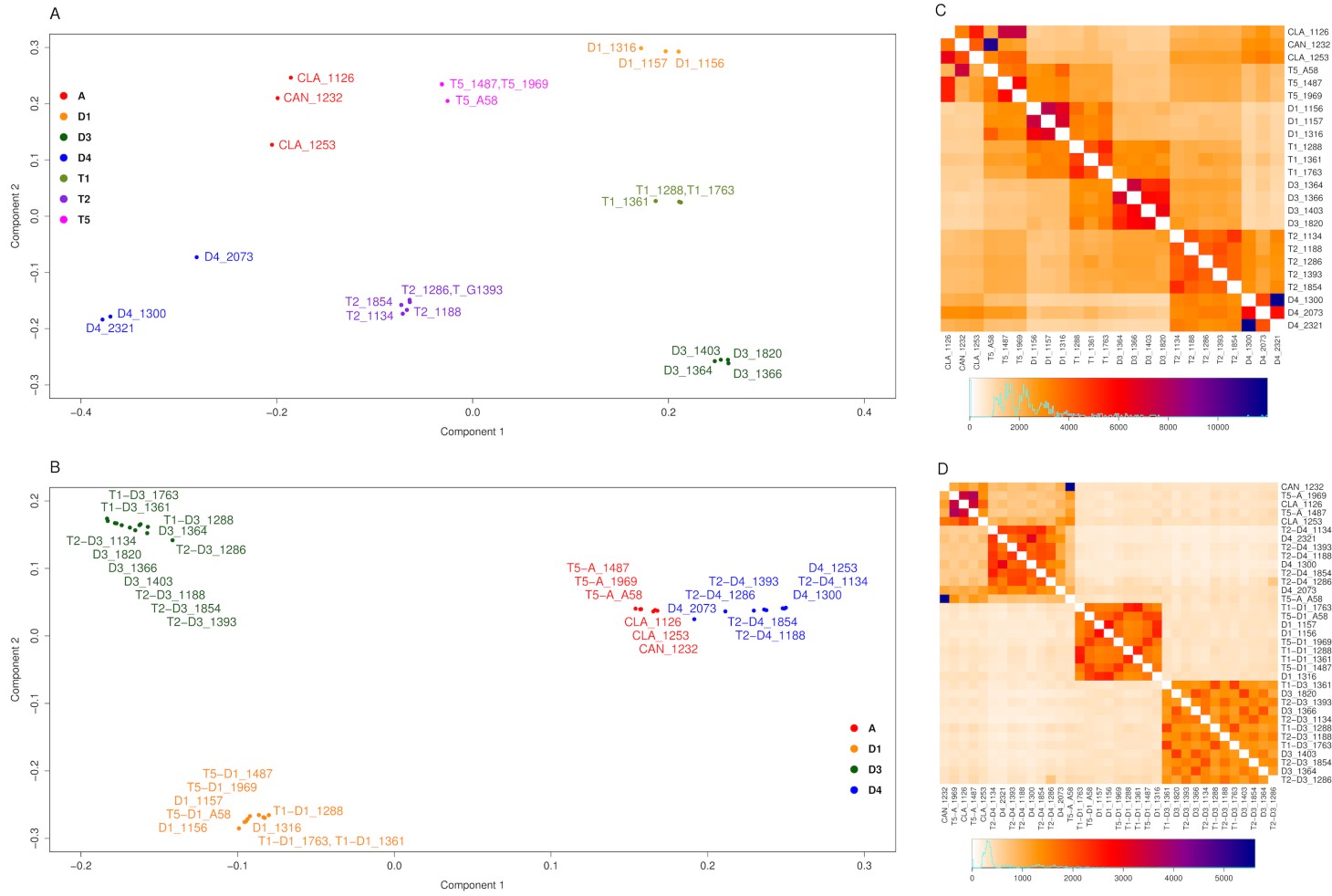

**Figure 4** **FineStructure analysis for *Glycine* perennial polyploid species.** Analysis using FineStructure for (A, C) 220,952 SNPs for the *Glycine* perennial polyploid complex (species groups A, D1, D3, D4, T1, T2 and T5) without homoeologue separation. 7 clusters can be distinguished (one per species group) in the PCA analysis where polyploids are admixtures of the diploid progenitor groups (A). The heatmap (C) shows diploid hybrid signal for polyploids, for example T5_A58 shows a stronger signal with its progenitors: CAN_1232 (blue) and D1_1316 (intense orange). (B, D) 70,910 SNPs for the *Glycine* perennial polyploid complex after homoeologue separation. 3 clusters can be distinguished in the PCA analysis (B): right cluster, species from the A-genome (A and D4); bottom-left cluster, species D1; and top-left cluster, species D3. (D) The heatmap signal is divided into the same three major clusters.

## Phylogeny and network analysis of concatenated SNPs

Phylogenetic and network analyses were conducted using the homoeologue dataset, with SNPs concatenated to create a single supermatrix. The maximum likelihood (ML) tree, rooted with *G. max*, identified four subclades comprising two major clades: (1) the A-genome, with subclades of D4 vs. *G. clandestina* and *G. canescens*; and (2) the D-genome (D3) and E-genome (D1) (Fig. 5A). Each of the subclades showed a different pattern with respect to diploid and tetraploid subgenome relationships. In the *canescens/clandestina* clade, the A-subgenome of the synthetic allopolyploid (A58) was sister to the accession from which it was created (*G. canescens* 1232), as expected, though with deeper coalescence than expected from an artificial hybrid; the two natural T5 allopolyploids were sister to

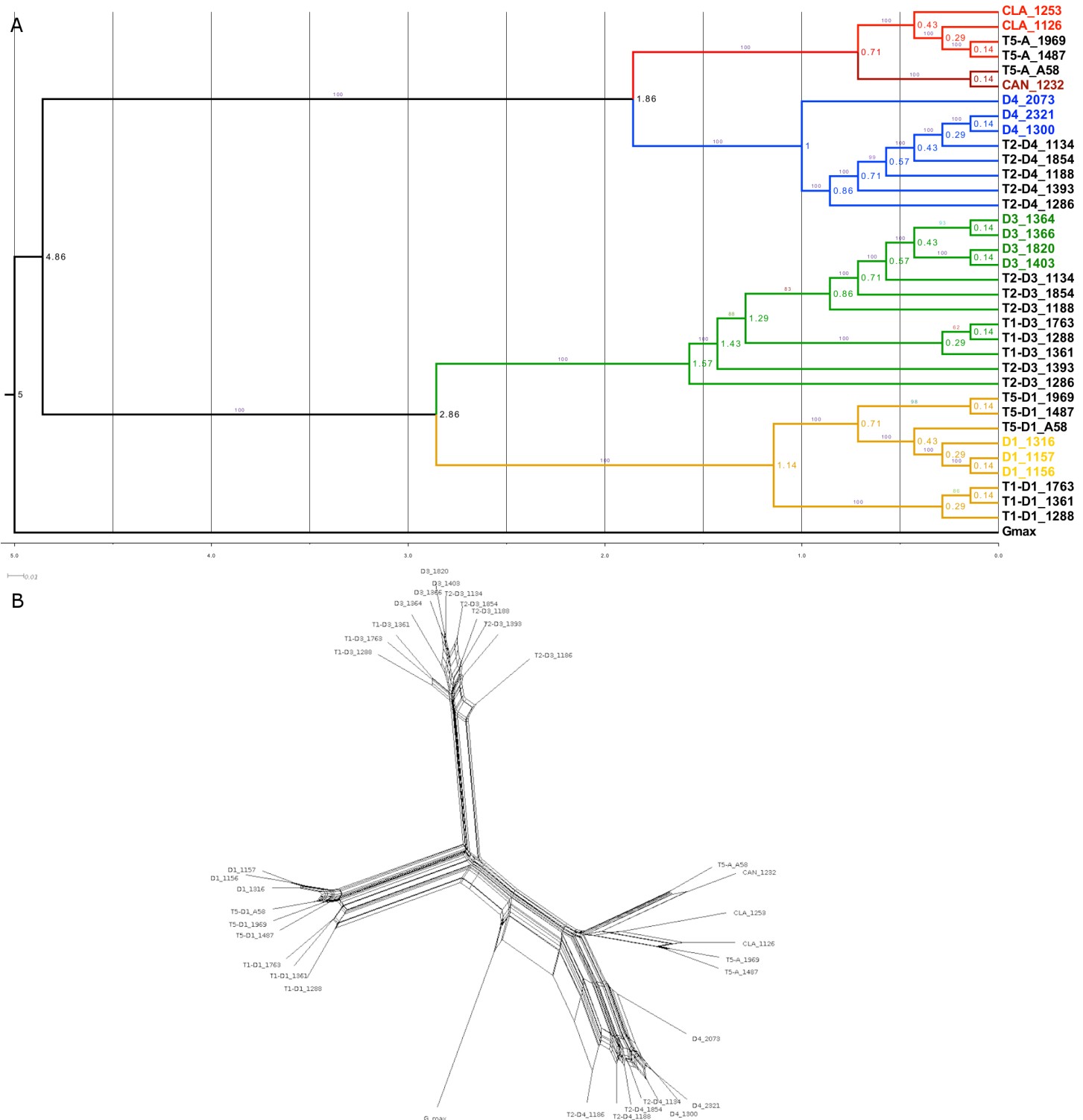

**Figure 5 Phylogenetic relationship in the *Glycine* perennial polyploid complex.** Relationships in the Glycine perennial polyploid complex after homoeologue separation, using a concatenated dataset. Branches are colored as in Fig. 4, based on the 5 different diploid species. In both the maximum likelihood (ML) phylogeny (A) and the NeighborNet network (B), the same major species groups are visible (D1, D3, D4 and *G. canescens/G. clandestina*).

*G. clandestina* 1126, as expected from other data (e.g., *Doyle et al., 2002*). In the D4 clade, diploid accession 2073 was sister to all remaining accessions, a unique placement consistent with its apparently admixed nature (Fig. 3A). The polyploid subgenomes formed a paraphyletic group, with the two diploid accessions sister to the D4 subgenome of one T2 accession (1134). A similar pattern was seen in the D3 subclade, where T2 accessions formed a paraphyletic group, and all four diploid accessions formed a clade sister to T2 accession 1134. Also embedded within the T2 accessions was a clade consisting solely of T1 accessions. T1 accessions also formed a monophyletic group within the D1 clade, where natural T5 accessions and D1 accessions also formed monophyletic groups. Surprisingly, there was not a sister relationship between the D1-subgenome of synthetic allopolyploid A58 and the D1 accession from which it was formed (1316). Similar topologies were produced by neighbor-joining analysis (data not shown).

NeighborNet was used to analyze the full homoeologue dataset to identify minority patterns of relationships in the data. When rooted with *G. max*, the topology (Fig. 5B) was very similar to the ML tree (Fig. 5A), even having such features as the sister relationship of D4 2073 to other D4 accessions, and the monophyly of T1 homoeologues in both the D1 and D3 clades. There was clear evidence of character support for alternative relationships, but those relationships were minor in comparison with the major phylogenetic signal.

## Gene-based phylogenetic and network analyses

Gene trees were constructed for the 27 genes (described in the Material and Methods) using several different phylogenetic and network methods. Similar topologies for trees from individual genes were obtained with BEAST and PhyML. All 27 trees showed the split between the A-genome clade and the D1/D3 clade seen in the ML tree reconstructed from concatenated SNPs (Fig. 5A). However, many individual gene trees showed unexpected groupings of one or more accessions, particularly within the A-genome clade, where several trees grouped accessions from *G. canescens* with *G. syndetika*-D4 instead of with *G. clandestina* (for example ML and BEAST trees for the gene Glyma04g39670, Figs. S17 and S45). Relationships within the major subclades varied among the 27 gene trees. For example, nine of the 27 trees showed separate clades for *G. canescens* (plus the A58 sequence) and *G. clandestina* (e.g., Figs. 6A and 6C), but in only three of them did diploid species form monophyletic groups (Figs. S13–S67). Overall, there were far more departures from expectations in the A-genome clade than in the D1/D3 clade.

There were numerous cases where alleles from diploid accessions formed monophyletic groups (e.g., 12 of 27 BEAST topologies had alleles from all four D3 accessions in a clade, often with high posterior probability). At some loci, alleles from one or more polyploids formed monophyletic clades; for example, at Glyma06g18640 (Fig. S50), all taxa, including both homoeologous subgenomes of each polyploid, formed separate clades, with the exception of *G. clandestina*. However, this was unusual, and paraphyletic groupings of alleles were common, particularly in polyploids. For example, at 26 of 27 loci, T2–D3 alleles were not monophyletic, at least some having closer relationships to D3 or T1–D3 alleles, and in gene Glyma01g35620, T5–D1_1969 was most closely related to D1_1156

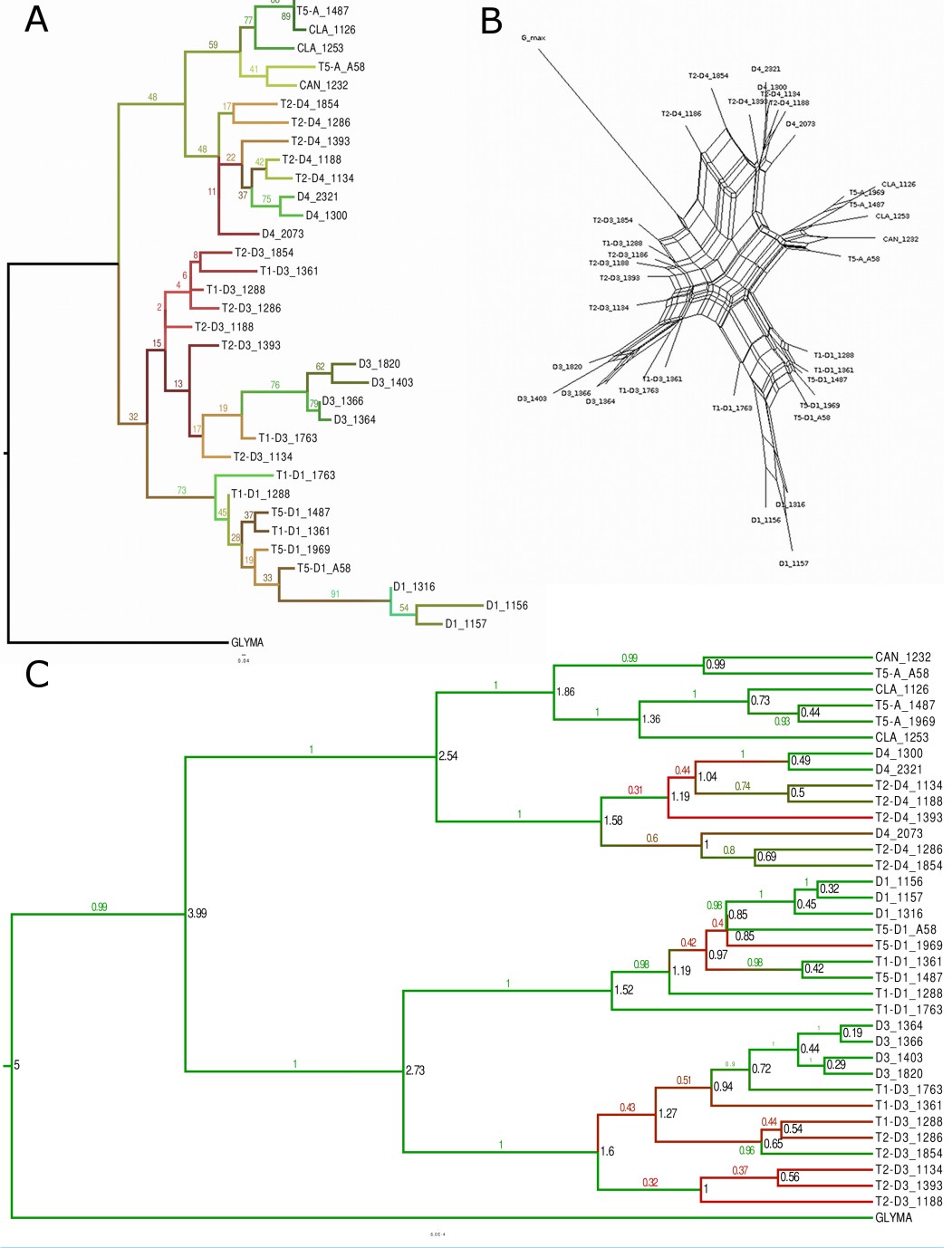

**Figure 6 Phylogenetic analysis for the Glyma02g11580 locus.** Glyma02g11580 locus using ML with bootstrap values (A), NeighborNet (B), and BEAST with posterior probabilities and showing node ages (in black) (C). For figures (A) and (C), branch length colors represent bootstrap or posterior probabilities values, with red shades being the lowest values and green shades being the highest values.

whereas T5–D1_1487 was most closely related to D1_1157 and D1_1316 (Fig. S41). On the assumption that alleles in tetraploids all originated from diploid progenitor species, such paraphyletic relationships suggest the input of alleles from different genotypes of diploid progenitors, due either to multiple origins or, alternatively, to continued gene flow from diploids after polyploid formation, perhaps involving unreduced gametes.

The BEAST trees, calibrated with the 5 MYA divergence of *G. max* and the perennial subgenus (*Innes et al., 2008*), allowed dates of allele divergence to be estimated. Among comparisons of interest are the minimum divergences between alleles from a tetraploid and alleles from its diploid progenitor (e.g., T2–D3 vs. D3) or alleles from the same progenitor in a second tetraploid (e.g., T2–D3 vs. T1–D3); the latter represent "diploid" alleles as well, under the assumption that there has been no gene flow between the two tetraploids, something that is reasonable for *G. tomentella* tetraploids (e.g., *Doyle et al., 1986*). Minimum distances between polyploid and diploid alleles (over)estimate the time of entry of that allele into the polyploid, which is typically assumed to be an origin of the polyploid (*Doyle & Egan, 2010*). Minimum dates (Table S3) were 0.31 MY for T1 (measured at the D1 locus), 0.29 MY for T5 (measured at the D1 locus), and 0.38 MY for T2 (measured at the D3 locus). Error bars on these estimates, however, were substantial.

NeighborNet (implemented in SplitsTree 4; *Huson & Bryant, 2006*) was used to construct networks for each of the 27 genes. Several networks showed patterns consistent with intragenic recombination; the Pairwise Homoplasy Index (PHI) of *Bruen, Philippe & Bryant (2006)*, also implemented in SplitsTree, was significant for 11 of the 27 genes (data not shown). The dominant patterns in NeighborNet topologies were similar to the overall pattern shown in phylogenetic analyses of the 27 genes, and thus to results for the full homoeologous SNP dataset. As with other methods, NeighborNet networks suggested multiple inputs of alleles from diploid progenitors into polyploids (e.g., gene Glyma02g11580, Fig. 6C).

## Species tree reconstruction under the coalescent

Species trees were reconstructed using the coalescent approach implemented in *BEAST (*Heled & Drummond, 2010*), which used information contained in the individual gene trees from the 27 genes described above. The overall *BEAST tree (Fig. 7A) topology was similar to that of trees from concatenated SNPs. By definition, each of the allopolyploid homoeologous genomes was a single OTU despite the possibility of independent origins; each of these was grouped with its putative progenitor species. Within the D1 genome clade, the T1 and T5 polyploids were sisters to one another; similarly, T1 and T2 were sisters in the D3 clade. The DensiTree output (Fig. S40) indicated considerable uncertainty only within the D3 clade, where both other possible topologies (T2 sister to D3, T1 sister to D3) appeared in a substantial number of trees. As expected, divergence dates of polyploids from their diploid progenitors estimated by *BEAST were higher than minimum estimates from the 27 individual loci, all being greater than 300,000 years (Fig. 7A).

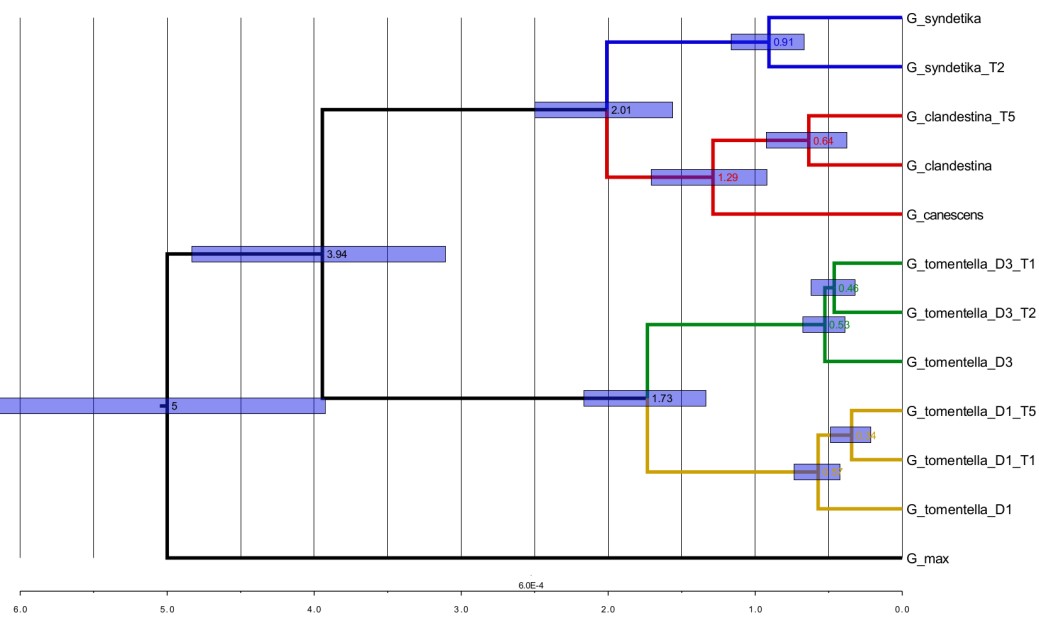

**Figure 7  Phylogenetic tree with estimated divergence dates.** *BEAST tree with the estimated node ages and error bars representing the highest posterior density (HPD) interval at the 95% level.

## DISCUSSION

The *Glycine* subgenus *Glycine* polyploid complex appears ideally suited as a model for studying allopolyploid evolution, because it comprises eight independently formed but closely related allopolyploid species triads (an allotetraploid and its two diploid progenitors; Fig. 1) that overlap in their genomic compositions. We are exploiting this model system to study the effect of allopolyploidy on a wide range of phenotypes, including transcriptome size, morphology, anatomy, climate niche, photosynthesis, and photoprotection (*Coate & Doyle, 2010*; *Coate et al., 2012*; *Ilut et al., 2012*; *Coate & Doyle, 2013*; *Coate et al., 2013*; *Hegarty et al., 2013*; *Coate, Bar & Doyle, 2014*; *Harbert, Brown & Doyle, 2014*).

To enhance the utility of this model group, it is important to move to a genome-wide understanding of their biology. As noted above, origins of the *Glycine* allopolyploids were hypothesized initially from crossing data and more recently from gene phylogenies, but inferences have been made from only two nuclear genes. Both of these markers supported the hypotheses of fixed hybridity of *Glycine* allopolyploid species. However, it is not known to what extent the entire genomes of these plants retain contributions from both parental diploid species in the face of potential loss due to initial genomic shock (*McClintock, 1984*), or other processes such as "genome downsizing" (*Leitch & Bennett, 2004*), fractionation (*Schnable & Freeling, 2011*; *Freeling et al., 2012*), or concerted evolution (e.g., *Wang et al., 2007*).

### Glycine allopolyploids are fixed hybrids throughout their genomes

Analyses using all SNPs identified from the full dataset showed that all three of these allopolyploids are indeed fixed hybrids, combining diploid genomes as depicted in Fig. 1.
Structure results indicated an essentially equal contribution from both parental diploids in all three cases (Fig. 3A); PCA analysis also was consistent with this hypothesis, placing each polyploid approximately midway between its putative progenitors, as expected for an F1 hybrid (Fig. 4A).

In order to determine whether or not the polyploids have contributions from their parents across their entire genomes, reads were partitioned by homoeologous genome and mapped to the soybean reference genome (*Schmutz et al., 2010*). As portrayed by e-chromosome painting (Fig. 2), it is clear that no individual sampled from any of the three allopolyploid species has any major regions represented by only one homoeologue. Coverage is sparse in pericentromeric and centromeric regions, as expected due to the low density of genes in these regions of the soybean genome (*Schmutz et al., 2010*). The degree of shared synteny between soybean and these perennial *Glycine* species is as yet unknown, but regardless of the order of chromosomal segments, it is clear that there has not been significant loss of homoeologous genes. We mapped reads to over 22,000 of the approximately 46,000 genes of the soybean genome (*Schmutz et al., 2010*). These numbers include both homoeologous copies from the 5–10 MYA polyploidy event that shaped the modern "diploid" ($2n = 38, 40$) *Glycine* genome. We were able to deconvolute between 4 and 19% of these 22,000 genes into their homoeologous contributions in each of the three recent allopolyploids (e.g., T1D1 and T1D3). Using genomic in situ hybridization (GISH), *Chester et al. (2012)* showed examples of allopolyploid *T. miscellus* plants that had all four chromosomes or chromosome segments of one diploid parent (4:0), but also examples of plants with 3:1 ratios of homoeologous chromosomes or chromosomal segments. Our e-chromosome painting method cannot distinguish the 3:1 condition from an equal contribution from both parents segments, so it is possible that such plants exist in our sample.

Structure analysis using the partitioned homoeologous SNPs corroborated results with the full, unpartitioned dataset, in placing each polyploid homoeologous genome with its putative progenitor (Fig. 3B). The FineStructure PCA supported three major groupings, each of which included diploids and the expected polyploid homoeologous subgenomes derived from them (Fig. 4B). The grouping of D4 accessions and two A-genome species (*G. canescens* and *G. clandestina*), along with polyploid genomes derived from them, into a single cluster is not surprising, because *G. syndetika* (D4) is also a member of the A-genome (*Ratnaparkhe, Singh & Doyle, 2011*). As noted above, genome groups were originally defined on the basis of reproductive compatibility in artificial crosses (*Ratnaparkhe, Singh & Doyle, 2011*), and indeed *G. syndetika* (D4) 2073 shows evidence of admixture with *G. canescens* and *G. clandestina* (Fig. 3). In contrast, D1 and D3, though both classified as "*G. tomentella*", belong to two different genome groups (E and D, respectively; *Ratnaparkhe, Singh & Doyle, 2011*). This greater genetic similarity of the three A-genome species is not reflected in relative divergence dates; for example, the *BEAST analysis dates the divergence between *G. syndetika* and the two other A-genome species at slightly earlier than the divergence between D1 and D3 (Fig. 7A). Thus, reproductive barriers likely arose earlier in the D1/D3 lineage than within the A-genome.

Allopolyploid evolution in *Glycine* fits "Darlington's Rule" (*Darlington, 1937*)—that allopolyploids should form between species that are reproductively isolated, often due to chromosomal differences, whereas reproductively compatible diploids instead tend to form homoploid hybrids. No allopolyploids are known to have formed among A-genome species, and only one of the eight known *Glycine* allopolyploids involves hybridization within a genome group (tetraploid *G. tabacina* is the product of the most divergent species cross possible within the B-genome; *Doyle et al., 2004*). D1 and D3, which as noted belong to different genome groups, have different chromosome numbers ($2n = 38$ vs. 40, respectively), which may contribute to their inability to form fertile diploid hybrids. D1 has also formed allopolyploids with D5A, another $2n = 40$ "*G. tomentella*"; however, reproductive incompatibility also occurs between $2n = 40$ *G. tomentella* taxa (*Doyle et al., 1986*), and other allopolyploids in the complex combine genomes of two $2n = 40$ taxa (Fig. 1).

## Gene histories, allele divergence times, and sources of genetic diversity in polyploids

Gene trees from the 27 loci were selected that met criteria designed to provide orthologues. These genes are highly transcribed with sufficient characters for phylogeny reconstruction, and inferences of polyploid origins mostly conformed to expectations based on previous work using the low copy nuclear locus, histone H3D (*Brown et al., 2002*; *Doyle et al., 2002*; *González-Orozco et al., 2012*), the nrDNA ITS (*Singh, Kim & Hymowitz, 2001*; *Rauscher, Doyle & Brown, 2004*), and chloroplast noncoding sequences (*Hsing et al., 2001*). The use of BEAST and *BEAST (*Heled & Drummond, 2010*) allowed us to estimate divergence times of alleles and species for the first time for some of these taxa. Dating polyploid origins is complicated by numerous factors (*Doyle & Egan, 2010*). For one thing, if the polyploid has arisen recurrently, then there is no single date that marks "the" origin. Given that polyploids are often invasive (e.g., *Pandit, Pocock & Kunin, 2011*), and the *Glycine tomentella* allopolyploids appear to be recently formed based on sharing identical histone H3D and nrDNA ITS alleles with their putative progenitors (*Doyle et al., 2002*; *Rauscher, Doyle & Brown, 2004*), we have speculated that they could have originated as a response to ecological disturbance due to human colonization of Australia, around 40,000 years ago (*Hudjashov et al., 2007*; *Pugach et al., 2013*). The relevant date for testing this anthropogenic disturbance hypothesis would be the oldest origin of each polyploid. However, because it is unlikely that a polyploid allele and any of a set of diploid progenitor alleles will coalesce at exactly the time of polyploid origin, distances for any given polyploid event will be overestimates of the actual time of origin. Further complicating matters, the error bars on our BEAST divergence estimates were large relative to the estimates themselves. Nevertheless, because even the minimum estimates of allele divergences between diploids and tetraploids are around 0.3 MY, it appears likely that these *G. tomentella* allopolyploids are hundreds of thousands rather than tens of thousands of years old. *BEAST estimates should be averages of all origins of a polyploid taxon, and these, too are several hundred thousand years for each allopolyploid. Thus, it appears likely

that these polyploid species were present in Australia before humans arrived there. The fact that these three species, and possibly other allopolyploid members of the complex, may have evolved at roughly the same time is intriguing. In the ca. 5 MY since the perennial members of *Glycine* diverged from the annual lineage (*Egan & Doyle, 2010*), there is no evidence of polyploidy until these species were formed, apparently well within the last 1 MY. Perhaps the onset of severe aridity in Australia around 3 MYA, heralding the change to the present extreme wet-dry glacial cycles (*Crisp, Cook & Steane, 2004*) could have provided ecological opportunities for polyploids. It will be interesting to refine our estimates through increased sampling of these three triads, and to obtain estimates for the other five allopolyploid species.

## ACKNOWLEDGEMENTS

The authors thank Sue Sherman-Broyles for helpful comments throughout the project. We also are grateful to Steven Cannon and a second, anonymous reviewer for detailed comments and suggestions.

### Funding

We received longstanding support from the US National Science Foundation for our research on Glycine, most recently awards 0822258 and 0939423. The funders had no role in study design, data collection and analysis, decision to publish, or preparation of the manuscript.

### Competing Interests

The authors declare there are no competing interests.

### Author Contributions

- Aureliano Bombarely conceived and designed the experiments, analyzed the data, wrote the paper, prepared figures and/or tables, reviewed drafts of the paper.
- Jeremy E. Coate conceived and designed the experiments, performed the experiments, contributed reagents/materials/analysis tools, reviewed drafts of the paper.
- Jeff J. Doyle conceived and designed the experiments, contributed reagents/materials/analysis tools, wrote the paper, reviewed drafts of the paper.

### DNA Deposition

The following information was supplied regarding the deposition of DNA sequences:

GenBank accession: SRP011928 for *Glycine dolichocarpa, G. syndetika* and *G. tomentella* D3 accession reads.

Sequence Read Archive accession: SRP038128 (*G. canescens, G. clandestina. G. tomentella* D1, T1 and T5).

## Supplemental Information

Supplemental information for this article can be found online at http://dx.doi.org/10.7717/peerj.391.

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
