# Peer review of "Mining transcriptomic data to study the origins and evolution of a plant allopolyploid complex"

_PeerJ, doi:10.7717/peerj.391_

## Round 0.1 · original submission · Minor Revisions

· Academic Editor

Minor Revisions

Your paper has now been seen by 2 reviewers, both of who think this paper could be of interest to the readership of PeerJ. Therefore, I would like to encourage the authors to submit a revised version in which they have carefully addressed all issues brought up by the reviewers, after which the paper could be accepted for publication.

·

Basic reporting

The basic reporting is sound.

Experimental design

The experimental design was sound, as were the methods. The approach was essentially: deep transcriptome sequencing of the species involved - which was used to construct transcriptome assemblies, and then to determine phylogenetic relationships among the diploid and tetraploid species and the subgenomes.

Validity of the findings

The findings look valid and reasonable to me.

Additional comments

I think this is a nicely done study and paper. I have a number of (mostly minor) comments and suggestions:

Abstract:
1. "Allopolyploidy ... is a common mechanism for producing new species..."
The wording "mechanism for producing" here looks a bit teleological. Allopolyploid is a process that often *results* in speciation.

2. "each of the three polyploid species are fixed hybrids combining the homoeologous genomes"
I am not sure that "homoeologous" adds anything here, since allopolyploids are always(?) composed of genomes with relatively similar chromosomes.

Introduction:
3. "All seed plants are fundamentally polyploidy"

4. "with a second WGD event characterizing all flowering plants" -- "characterizing" doesn't seem the intended word here.

5. "This Glycine-specific WGD occurred around 10 MYA ... and 5 MYA ..."
- is there a missing "between" in this sentence? Otherwise, the two dates are hard to parse.

6. "it is not known to what extent the entire genomes of these plants retains contributions" -- it looks like "retains" should be made singular.

Methods:
7. "joined with the Cat Linux command" -- this should probably be "joined with the 'cat' Linux command" (if it is even necessary to give the name of the command; maybe just say "were concatenated")

Results:
8. "Between ~350 and ~1,350 full length sequences were assembled for the polyploid homoeologous subtranscriptomes of which between 4 to 19% were duplicated genes from the last Glycine WGD event"
-- I don't understand this: "of which between 4 to 19% were duplicated genes from the last Glycine WGD event." Shouldn't most of the transcript assemblies have had identifiable homoeologs?

Discussion:
9. Somewhere when the idea of "multiple origins" is mentioned (as it is in the Abstract, Introduction, Results, and Discussion), it would be helpful to me to see a more explicit description of what this means in this context. Strictly, I suppose any individual derives from a long lineage of "single origins" (i.e. a succession of single crosses - either wide or narrow). So exactly what does "multiple origins" mean? Does it apply only at a population level? I suppose in this context it means multiple independent polyploidies involving the same parents (e.g. D3 and D4), giving rise to several (similar) allopolyploid progenitor species (e.g. T2a, T2b), at different times -- followed by subsequent crossing of those allopolyploids, to produce a new "allopolyploid hybrid." Precisely how is this be seen in the analyses in this paper? I believe I see this pretty clearly in Figure 5 A -- but I needed to work at it. I think the closest the paper comes to such a description is around the sentence "such paraphyletic relationships suggest multiple origins or perhaps subsequent gene flow from diploids after polyploid formation" -- but I have to admit that this paragraph is difficult for me to digest. I think this is partly due to potential ambiguity in the terms "monophyletic" and "paraphyletic" in this context - do these refer to the subgenomes ... which are sometimes components of diploids, and sometimes of tetraploids? This is muddied by the muddiness of the "species" concept in this complex. As a reader, I need to internalize a pretty complex phylogenetic model before unpacking and making sense of "paraphyletic." This (important) paragraph is also trying to make several other points. Some simplification and separation would be helpful.

10. (just speculation): I wonder: do the 2n=38 or 2n=78 accessions show evidence of chromosome loss? If not, that probably implies chromosome fusion. Would such a fusion be evident either with GISH or other cytological methods? Are there hints of this from previous studies? This might be worth a brief discussion - in light of the chromosomal losses observed in e.g. Tragopogon.

Figures:

11. Figure 6: what is the meaning of the branch length colors?
Can the fonts on the trees (especially the posterior probabilities) be made larger - if necessary, by exporting to a vector program and resizing?

Reviewer 2 ·

Basic reporting

This manuscript utilizes a massive sequence data set gathered for another experiment/manuscript to investigate (1) whether or not previous reports on the origins of a group of perennial Glycine allopoyploid species based on a few nuclear loci were accurate and (2) if there is genome-wide divergence of signal / evidence of genomic loss following polyploidy. After a fairly exhaustive analysis, the short answers appear to be yes and no, respectively. The methods are generally appropriate and the writing is fine; however, there are numerous issues that require correction, clarification, or reconsideration. These issues are line-itemed in the following sections.

Experimental design

Essentially fine but there are places where more detail is required on the analysis methods; these are detailed in the General Comments section.

Validity of the findings

Essentially fine but there are places where more detail is required on the analysis methods; these are detailed in the General Comments section.

Additional comments

The introduction does not describe the species under study very well. Information in the first paragraph of the Methods and second paragraph of the Discussion (starting on ln-346) should be moved to the introduction. Also, how the A, D, E genomes have been defined historically should be clarified in the Introduction.

It would also be helpful to introduce the specific questions of interest in the Introduction. At present, this doesn’t really happen until the fourth paragraph of the Discussion.

The synthetic lineage is variously referred to as A58, A58-1, and A-58. One lineage name should be used throughout.

The synthetic lineage (A58) is not introduced in the Methods section but should be.

Ln-39 – “around 10 MYA” should probably read “estimated between 10 MYA”. Currently reads as though there were 2 events, one at 10MYA and another at 5MYA.

Ln-25 – origin should read “origins”

Ln-52 – origin should read “origins”

Ln-65 – lists G. clandestina as putative parent of T5 but Figure 1 lumps G. glandestina with G. canescens and both are analyzed. This should be resolved/explained in the Methods section.

Ln-86 “ perennial species reads were processed . . .”. Were there annuals included in the study? Up to this point in the ms, there is no mention/designation of annual species in the study sample.

Ln-103 – rebuilt should read “rebuild”

Ln-104 – sentence beginning on this page is incomplete.

Ln-107 – The section describing the Structure analyses needs more detail, including:

- how many iterations of data collection (burn-in rep’s is reported but not data collection)

- recent reports suggest 100,000 generations of burn-in. Might be worth repeating some runs to verify results with longer burn-in.

- how was Lambda calculated?

- was admixture selected or not? See later comments on plots.

It is often difficult to identify precisely what a set of OTUs or accessions consists of for a particular set of analyses – a Table that details information (e.g., taxa, sampling, homoeolog sets, etc.) that could be referenced in the Methods would be very helpful.

Ln-167 – again, “perennials” are referred to but still haven’t seen the distinction between perennials and annuals in the data sets. Or is this just a general reference to the “perennial Glycine polyploids”? If so, then I would suggest simply introducing the system in the Introduction and Methods as such and stop referring to the “perennials” throughout as it is confusing.

Ln-345 – sentence needs a period.

Ln-365 – sentence beginning on this line is not clear.

Ln-371 – “unequal contributions from its two diploid progenitors” is probably not the most accurate phrase. It is very likely that there were equal genomic contributions to the Trag polyploids but that there has been differential loss.

Ln-400 – the Chester reference needs its parentheses adjusted.

Ln-423 – the sentence beginning on this line is confusing. Different chromosome numbers contributes to forming fertile hybrids? Explain.

Ln-426 – incomplete sentence

Ln-474 – Darlington ref. is incomplete

Ln-488 – incomplete reference

Ln-513 – incomplete reference

Figure 1 legend – Glycine needs to be italicized.

Figure 1 legend - “. . . allotetraploid species by circles.” Should read “. . . allotetraploid species by squares.”

Figure 1 legend – number should read “numbers”.

Figure 2 legend – first line is repeated

Figure 3 and Results and Discussion – In several places, the degree/absence of admixture is dicussed but the Structure plots do not look like plots resulting from runs with “Admixture” selected and the Methods don’t include that information. This should be clarified. If “No admixture” was selected, then the discussion around the results with reference to admixture as a biological process might be re-considered. Alternatively, the results may be robust enough that this setting does not have much effect. The authors may consider rerunning a subset (focusing on relevant K values) to confirm the findings.

Figure 4 legend – “heapmap” should read heatmap

Table 1 – Glycine max should be italicized.

---

## Round 0.2 · accepted · Accept

· Academic Editor

Accept

The authors have addressed all issues brought up by the reviewers and this paper can nog get accepted for publication in PeerJ.